# Children's understanding of when a person's confidence and hesitancy is a cue to their credibility

Susan A. J. Birch[1]*, Rachel L. Severson[2]*, Adam Baimel[3]

**1** Department of Psychology, University of British Columbia, Vancouver, British Columbia, Canada,
**2** Department of Psychology, University of Montana, Missoula, Montana, United States of America,
**3** Department of Psychology, Health and Professional Development, Oxford Brookes University, Oxford, United Kingdom

\* sbirch@psych.ubc.ca (SB); rachel.severson@umontana.edu (RS)

**Data Availability Statement:** The data underlying this study have been deposited to the OSF database at https://osf.io/ugczm/. DOI: 10.17605/OSF.IO/UGCZM.

## Abstract

The most readily-observable and influential cue to one's credibility is their confidence. Although one's confidence correlates with knowledge, one should not always trust confident sources or disregard hesitant ones. Three experiments ($N$ = 662; 3- to 12-year-olds) examined the developmental trajectory of children's understanding of 'calibration': whether a person's confidence or hesitancy correlates with their knowledge. Experiments 1 and 2 provide evidence that children use a person's history of calibration to guide their learning. Experiments 2 and 3 revealed a developmental progression in calibration understanding: Children preferred a well-calibrated over a miscalibrated confident person by around 4 years, whereas even 7- to 8-year-olds were insensitive to calibration in hesitant people. The widespread implications for social learning, impression formation, and social cognition are discussed.

## Introduction

*Confidence is good, but overconfidence always sinks the ship. ~ Oscar Wilde*

The human proclivity to learn from others has captured the attention of scholars from a wide range of disciplines. Justifiably so! Person-to-person transmission of information allows us to learn infinitely more than if we were restricted to learning first-hand through time-consuming trial-and-error. It also enables the uniquely human capacity for 'cumulative cultural transmission'—the transmission of information (social, cultural, scientific, etc.) from one generation to the next [1]. Some information is impractical or dangerous to acquire first-hand (e.g., which berries are safe to eat), whereas other information is impossible to learn first-hand (e.g., language, history). Consequently, the vast majority of the information humans acquire will come from others.

Of course, learning from others comes with many challenges: People differ in their level of intelligence and their areas of expertise. They frequently offer their opinions rather than facts.

**Funding:** This research was funded by a grant from the Social Sciences and Humanities Research Council of Canada (http://www.sshrc-crsh.gc.ca) to SB (Grant No. 435-2013-0445). The funders had no role in study design, data collection and analysis, decision to publish, or preparation of the manuscript.

**Competing interests:** The authors have declared that no competing interests exist.

They can intentionally lie, and they can unintentionally convey misinformation out of ignorance or uncertainty. For these reasons, it is exceedingly important for learners to assess if (or when) others are providing credible information. Fortunately, there are several cues people use to decide whether someone is providing credible information, such as a person's professed level of knowledge or ignorance [e.g., 2, 3], area of expertise [e.g., 4], prior track-record of accuracy [e.g., 5, 6], perceptual access to information [e.g., 7], or a person's status or prestige [8; for reviews see 1, 9–14]. Another useful credibility cue (the focus of this manuscript) is a person's expression of confidence or uncertainty. Expressions of one's confidence or uncertainty can include verbal (linguistic) cues (e.g., saying 'I think' vs. 'I know'), nonverbal cues such as facial expressions, body language, and gestures (e.g., shoulder shrugging), and paralinguistic cues such as the intonation, volume, or rate of speech. Herein, when we use the word 'confidence' we are referring to the mental *state* that is context-dependent (e.g., Sally is confident in her answer) rather than confidence as a personality *trait* that persists across contexts (e.g., Sally is a confident person), unless specified.

Importantly, a person's expression of confidence is far from a perfect correlate of credibility. Nonetheless, research shows that a person's level of confidence in their statements and actions tends to co-vary with their knowledge [15]. And, for adults at least, the degree of confidence or uncertainty displayed by another individual is one of the most readily observable and commonly used cues to a person's credibility. For example, adults are more inclined to believe someone who appears confident than someone who appears uncertain [16], and confident individuals tend to be regarded as more influential in group settings [17]. Not only are adults more likely to learn from confident than uncertain people, they also learn faster and better from them [18]. This tendency to use another person's level of confidence as a cue to their credibility is sometimes referred to as the 'confidence heuristic' [19, 20].

The Confidence Heuristic Model maintains that people make rapid judgments of credibility, such as how accurate the information is or how trustworthy an individual is, based on the confidence expressed *in the message* or *by the messenger*. The confidence heuristic has been the focus of considerable attention in legal settings, political decision-making, clinical settings, and consumer economics [e.g., 21–25]. For example, adults tend to vote for more confident political candidates [26, 27], and are more trusting of testimony from confident witnesses in mock jury trials [28, 29].

In contrast to the wealth of literature on adult's sensitivity to a person's level of confidence, comparatively little research has examined these issues developmentally. The research that has been done suggests that young children can also use an individual's level of confidence to determine who might be a credible source of knowledge. For example, 4-year-olds reliably distinguish between verbal markers of certainty ("know") and uncertainty ("think" or "guess") [30], and prefer to learn new information from someone who expresses certainty in their knowledge rather than uncertainty [2, 31]. Even children as young as two years of age preferentially imitate the actions of individuals who display nonverbal cues of confidence over those who appear uncertain [32, 33].

Clearly, confidence is a compelling cue of knowledge and trustworthiness, even for young children. However, the level of confidence people display is their own *subjective* assessment of what they know, and therefore prone to considerable error. Sometimes one's confidence is unjustified (i.e., they are confident when they are not knowledgeable—they are *overconfident*). Individuals can be overconfident in what they know because they are uninformed, misinformed, or 'faking' it. Moreover, some individuals are characteristically prone to overconfidence (i.e., they tend to be overconfident across many instances and contexts) [34]. As Oscar Wilde's quote above suggests, one should be wary of such unjustified confidence. In fact, a repeated history of being overconfident can signal *in*competence and a *lack of* credibility.

Importantly, as pointed out by Tenney, Spellman, & MacCoun [35], a more reliable credibility cue than confidence itself is the correlation, or 'calibration', between one's confidence and one's accuracy. Calibration refers to how well one's degree of confidence predicts one's likelihood of being correct. Well-calibrated individuals are confident when they are informed or accurate, and hesitant when they are uninformed or inaccurate; whereas miscalibrated individuals' degree of confidence is unrelated (high or low confidence regardless of knowledge) or negatively related to their knowledge (high confidence when ignorant, low confidence when knowledgeable).

In an inaugural experiment examining whether children are sensitive to a model's calibration when deciding whose testimony to trust, Tenney and colleagues [29] presented 5- and 6-year olds (as well as adults) with stories involving conflicting statements from two 'eye witnesses' to an event in which someone hit a ball through a window. One witness confidently claimed the perpetrator was "Tyler", whereas the other witness confidently claimed it was "Kenny." In addition, the witnesses made two other claims about the event: one about the weather and another about the color of the ball. Both witnesses confidently claimed it was sunny, whereas one witness was also confident about the color of the ball and the other was uncertain. That is, one witness stated, "I *know* it was sunny, I *know* the ball was red, and I *know* that Tyler hit the ball right through the window!" The other witness stated, "I *know* it was sunny, I *think maybe* the ball was blue, and I *know* that Kenny hit the ball right through the window!". When asked which witness they believed, both adults and children were more likely to favor the witness that was confident in all of her statements. Next, participants were told about the actual veracity of the witnesses' statements about the weather (both were correct: it was sunny) and the color of the ball (neither witness was correct: the ball was actually white) and were again asked whose accusation they believed. Armed with this additional information —that one witness was miscalibrated by being overconfident in one of her claims (that the ball was red), whereas the other was well-calibrated (i.e. she was confident when correct about the weather, and hesitant when incorrect about the ball's color)—adults now favored the testimony of the well-calibrated model. The 5- and 6-year-olds, on the other hand, continued to favor the testimony of the witness who made all claims with confidence suggesting they were insensitive to calibration as a cue to one's credibility. A follow-up experiment replicated 5- and 6-year-olds' insensitivity to calibration when learning new word labels [29; Experiment 1b] and another follow-up experiment showed that adults were similarly insensitive to the witnesses' calibration when under cognitive load [29; Experiment 4]. That is, adults under cognitive load, like the young children, preferred to believe the statements of the person who was the most confident *overall*, revealing how salient and influential a person's confidence is (even if sometimes unreliable).

Interestingly, young children are not *always* swayed by a person's confidence. Brosseau-Liard, Cassels, and Birch [36] examined which credibility cue–a person's prior accuracy or level of confidence–preschoolers find most compelling when learning from others. In their Experiment, 4- and 5-year-old children were presented with simultaneous information about the models' prior accuracy and level of confidence. In the history phase of their experiment, one model was confident when making obviously inaccurate claims (e.g., "Whales live in the ground!"), whereas the other model was hesitant when make accurate claims about those same facts (e.g., "umm. . .Whales live in the. . .ah. . .water?"). Then, during the test phase both models provided novel information with the same level of confidence (or lack thereof) they displayed with the previous facts (e.g., the previously accurate model would hesitantly say "I think that's a lanternfish?") and the previously inaccurate model would confidently say "I think that's a paddlefish!"). In choosing whose information to endorse (e.g., "Do you think it's a lanternfish or a paddlefish?"), 4-year-olds were at chance, whereas five-year-olds significantly

preferred the less confident (but previously accurate) model. Thus, when these two credibility cues are presented concurrently but are in conflict, 5-year-olds give more credence to one's prior accuracy than degree of confidence. These results are not in conflict with Tenney et al.'s [29] findings because in Brosseau-Liard et al. [36] the primary research question was whether preschoolers found a model's confidence or a model's prior accuracy more compelling, as such *both* models were miscalibrated (i.e., hesitantly accurate or confidently inaccurate).

Here, we expand on the work of Tenney et al. [29] and Brosseau-Liard et al. [36] to examine the development of children's understanding of the relationship between the confidence or hesitancy expressed in one's statements and the credibility of the message and the messenger. More specifically, this research addresses several open questions about how children's learning and impression formation is influenced by a model's confidence and hesitancy: 1) At what age do children begin to use calibration as an indication of a person's credibility, and does it change across development? 2) Does an individual's calibration influence children's learning decisions? And 3) Does an individual's calibration influence children's impression of how smart that individual is more generally? We investigated these questions across three experiments. In Experiment 1, we tested a large sample of children ranging in age from 3 to 12 to gauge the developmental onset and trajectory of a sensitivity to calibration. Specifically, when do children perceive a well-calibrated model (i.e., hesitant when uninformed) as more credible than a miscalibrated model (e.g., confident when uninformed)? In Experiment 2 we equated the models' level of confidence (i.e., both models confident) to isolate when they understand that a model's *confidence* can be unjustified rendering him or her less credible. Whereas, in Experiment 3 we equated the models' level of confidence (i.e., both models *hesitant*) to isolate when they understand that a model's *hesitancy* can be justified, rendering him or her more credible. Across all studies, we assessed participants' 1) preference of which model to learn novel information from, and 2) judgments of which model is 'smarter'.

## Experiment 1

Experiment 1 investigated at what age children appreciate that individuals who are better calibrated are better sources of information. In particular, we tested whether children (3–12 years) would (a) judge a well-calibrated model as 'smarter' than a miscalibrated model and (b) prefer to learn from a well-calibrated model. To do so, we first presented children with videos to establish that one model had a track-record, or history, of being well-calibrated and the other model had a history of being miscalibrated (i.e., the 'History Phase'). In the videos a male actor, Don, presented two adult female models with four covered boxes (that participants were told contained different pictures). For each box Don asked each model, "Do you know what is inside this box?" In response, one model confidently claimed to know the contents of each box (e.g., "It's a rabbit. I know it's a rabbit. It's a rabbit for sure!"), whereas the other model hesitantly offered an answer (e.g., "Umm, It could be a puppy. . ..hmm. . .maybe a puppy? I'll guess a puppy?"). Critically, in the Informed Condition both models were shown the contents of the boxes, whereas in the Uninformed Condition neither model saw inside the boxes. Consequently, the Confident model was well-calibrated in the Informed condition (i.e., her confidence was justified because she could see inside). In contrast, the Confident model was miscalibrated in the Uninformed Condition (i.e., her confidence was not justified because she did not see inside the boxes). The contents of the boxes were not visible to participants. Instead, our design capitalized on children's understanding of visual access (i.e., looking usually leads to knowing) that is evident in 3- and 4-year-olds [37, 38] and possibly even in infancy [39, 40].

**Table 1. Design overview and predictions for each experiment.**

| Exp. | Condition | History Phase | Test Phases (*Both have Visual Access*) | Prediction (if sensitive to calibration) |
|---|---|---|---|---|
| 1 | Informed (Baseline[a]) | 1 Model Confident-WC<br>1 Model Hesitant-MC | Both Confident | Test: Favor Confident |
| | Uninformed | 1 Model Confident-MC<br>1 Model Hesitant-WC | Both Confident | Test: Favor Hesitant |
| 2 | Both Confident | 1 Model Informed-WC<br>1 Model Uninformed-MC | Both Confident | History: Favor Informed Test: Favor Previously Informed |
| 3 | Both Hesitant | 1 Model Informed-MC<br>1 Model Uninformed-WC | Both Confident | History: Favor Informed Test: Favor Previously Uninformed |

WC = well-calibrated model; MC = miscalibrated model

[a] A preference for the well-calibrated model in this condition could be a simple preference for confidence; this is our baseline to ensure that in this design children are sensitive to confidence.

After establishing that one model was better-calibrated than the other during the History Phase, we assessed which model participants preferred to learn *new information* from (referred to as 'future learning') and who they thought was smarter. Previous research suggests that children generally believe that a confident model is smarter than a hesitant one [32, 36]. Here, the critical questions were whether participants in the Uninformed Condition would recognize that there is no basis for the Confident model's confidence (because she is uninformed and therefore miscalibrated) and instead prefer to learn new information from the Hesitant (well-calibrated) model, and also judge the Hesitant model as 'smarter'.

Given the work by Tenney et al. [29] and Brosseau-Liard et al. [36] discussed above, we predicted that younger children (e.g., 3- and 4-year-olds) would be swayed by confidence and would prefer to learn from, and judge as smarter, whomever appeared the most confident regardless of condition. However, we expected that older children would be warier of information from the Confident model when her confidence was not justified (Uninformed Condition) compared to when her confidence was justified (Informed Condition). In other words, we predicted that when the models were visually informed (Informed Condition), all children would prefer to learn from the Confident model and judge her as 'smarter'. However, when the models had no visual access to the information (Uninformed Condition), younger children would show a preference for the Confident model (who is miscalibrated), whereas older children would perceive the Hesitant model (who is well-calibrated) as smarter and prefer to learn new information from her. See Table 1 for a Design Overview and Predictions.

## Method

**Participants.** Participants included 502 children (50.2% female) ranging in age from 3 through 12 years (*M* = 6.42 years, *SD* = 2.36, range = 3.0–12.83; see S1 Table for full details). The child's age was treated as a continuous variable for regression analyses. Age was also analyzed categorically using the following age groups: 3–4 years (*n* = 162, 50% female, *M* = 3.93 years, *SD* = .56), 5–6 years (*n* = 148, 49.3% female, *M* = 5.97 years, *SD* = .55), and 7 years and older (*n* = 191, 51.3% female, *M* = 8.88 years, *SD* = 1.62). Participants' parents (55.6%) reported their child's ethnicity as Caucasian (61.3%), Asian (23.3%), mixed ethnicities (11.1%), and other ethnicities (4.3%) such as Aboriginal, African, or Arab descent. Participants were recruited through the Psychology Department's child participant pool (*n* = 218) and at a local science museum (*n* = 284). An additional 53 participants were excluded due to a failure to complete the procedure (*n* = 13), experimenter error (*n* = 8), parental interference (*n* = 2), language comprehension issues (*n* = 1), or because the parents reported a diagnosed or suspected

developmental delay (*n* = 29). The study procedure was approved by the Behavioral Research Ethics Board in the Office of Research Services at University of British Columbia. Parents of participating children provided written consent and children provided verbal assent to participate. These data were collected between February 2014 and June 2015.

**Materials and procedure.** Participants were randomly assigned to either the Informed (*n* = 241) or Uninformed (*n* = 261) condition. Each child was tested individually by a female research assistant in a quiet room at the university laboratory or museum. The session took approximately 10 minutes. The research assistant began by saying, "Hi (child's name)! My name is __. I have a fun game I'd like to play with you. Will you play with me? Great! We're going to watch some videos and I'm going to ask you some questions." Then, starting the videos for the History Phase and revealing photographs of two females in the videos saying, "Don is going to play a game with Andrea and Beverly. This is Andrea and this is Beverly. Don has some boxes and he's put a different picture inside each box. Let's watch!" Participants watched as the video depicted two female models who differed in their demonstration of confidence (i.e., confident or hesitant). A male actor (Don) presented both models with four covered boxes and, depending on the condition, were either shown (Informed condition) or not shown (Uninformed condition) the contents of the boxes (a picture of a familiar animal) (see Fig 1).

For each box, the male actor asked the models (a) whether or not they had seen inside the box ("Have you seen inside this box?") to which the models correctly responded "yes" or "no" consistent with the condition; and (b) to name what animal picture was inside the box ("What's inside this box?"). For each box, the Confident and Hesitant models provided conflicting answers regarding the boxes' contents (e.g., rabbit vs. puppy). Across conditions, when asked to state the boxes' contents, the Confident model provided non-verbal (e.g., head nodding), paralinguistic (e.g., declarative tone), and verbal cues of confidence (e.g., "It's a rabbit. I know it's a rabbit. It's a rabbit for sure."), whereas the Hesitant model provided non-verbal (e.g., shrugging shoulders), paralinguistic (e.g., questioning intonation, slower rate of speech), and verbal cues of uncertainty (e.g., "Umm, It could be a puppy. . ..hmm. . .maybe a puppy? I'll guess a puppy?"). Thus, in the Informed Condition, the Confident model was well-calibrated to the situation (i.e., she was knowledgeable of the boxes' contents and was confident in her answer), whereas the Hesitant model was miscalibrated (i.e., she was knowledgeable of the boxes' contents but expressed uncertainty in her answer). Conversely, in the Uninformed Condition, the Confident model was miscalibrated (i.e., she had no knowledge of

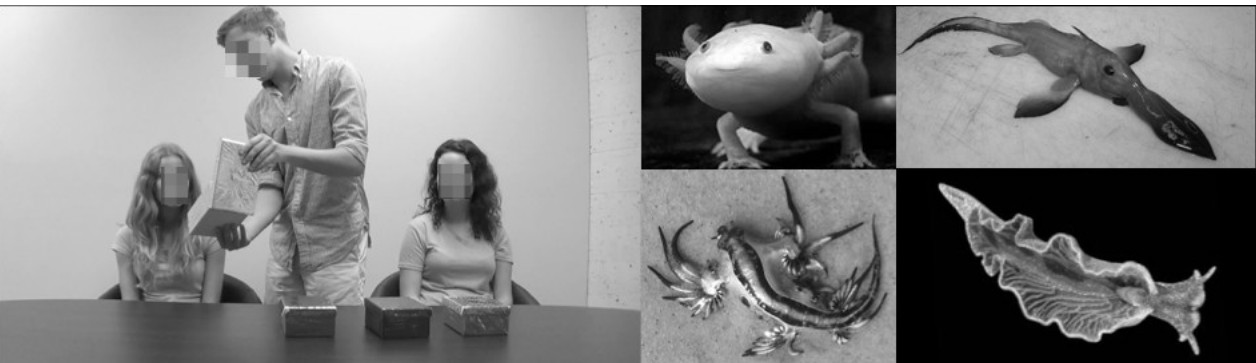

**Fig 1. Static image from the stimuli videos used during the History Phase (Left) and photographs of the novel animals used in the Endorse Test Phase (Right).** The research assistant models in this image have given written informed consent to use these images for research and publication; model names are pseudonyms.

the boxes' contents and thus should not be confident in her answer), whereas the Hesitant model was well-calibrated (i.e., she did not possess knowledge of the boxes' contents and was therefore uncertain of her answer). To further demonstrate that the Confident model in the Uninformed Condition was inaccurate and did not somehow have 'insider information' about the contents of the boxes to justify her confidence, the researcher told participants that both models were incorrect and then provided the correct answer (e.g., "It was actually a picture of a turtle."). Initial piloting with children did not include this check on 'insider information' and during post-test questioning participants generated reasons the confident model could know the boxes' contents (e.g., "She must have peeked inside before"). Indeed, attributions of privileged knowledge are prevalent among 3-year-olds but diminish by age 5 [41]. To correct for these assumptions of privileged knowledge we provided evidence that the Confident and Hesitant models both lacked knowledge of the boxes' contents and their answers were incorrect.

Two test phases followed: the Endorse phase and the Ask Phase. First, in the Endorse Test Phase, both models were shown a picture of an "unusual animal" (outside of the participant's view) and were asked to name the animal. Both models answered confidently but provided different novel names for the pictured animal (e.g., "I think it's a Modi/Toma. Yep, it's a Modi/Toma. I know it's a Modi/Toma!"). Although children as young as 4 years trust a source who uses the term 'I know' over 'I think' [30], this does not necessarily imply that children equate use of 'I think' with hesitancy, only that 'think' is less convincing than 'know'. In the Endorse Phase, both models used both terms in their responses. Additionally, the word 'think' can also be used to highlight that one's statement does, or might, conflict with another's response (e.g., You think it's X, but I think it's Y; as was the case in the Endorse trials). Participants were then asked to endorse one of the novel animal names provided by the models ("What do you think it's a picture of? A Modi or Toma?"). Participants could respond verbally or by pointing to the photograph of the model. This was repeated across four trials. Then, in the Ask Phase, participants were shown a picture of an unusual animal (i.e., one they had not previously seen; Fig 1B) and were asked: "Who do you want to ask what animal this is? Who do you think would know that?" This was repeated across four trials. The Endorse Phase always preceded the Ask Phase to demonstrate that the previously hesitant model was not *always* hesitant (i.e., she made confident statements in the Endorse Phase when she could see the objects) but was hesitant when she lacked knowledge (i.e., well-calibrated). Model role (Confident/Hestitant), speaking order (i.e., which model answered first), and which novel names were provided by the Confident model were counterbalanced. See Fig 2 for a visual schematic of the methods.

Finally, participants were asked a series of post-test questions. Post-test items 1–3 assessed whether participants understood and could accurately recall whether or not the models had seen inside the boxes ("Did Andrea and Beverly get to see inside?"), which model was confident ("Which one said they knew for sure what was inside? Andrea or Beverly?"), and which was hesitant ("Which one said they were guessing what was inside? Andrea or Beverly?"). Post-test item 4 assessed whether children recognized that one model was more credible ("Who do you think is smarter? Who knows more?").

## Results

**Preliminary analyses.** Preliminary analyses indicated that there were no effects on learning preferences of participant sex, data collection location, model speaking order, or novel answers provided (all $ps > .17$), thus all subsequent analyses were collapsed across these variables. However, an unexpected significant main effect of the identity of the model ($p < .01$) was found, and addressed below. Importantly there was no interaction between model identity and condition ($p > .35$).

Experiment 1: Informed Condition

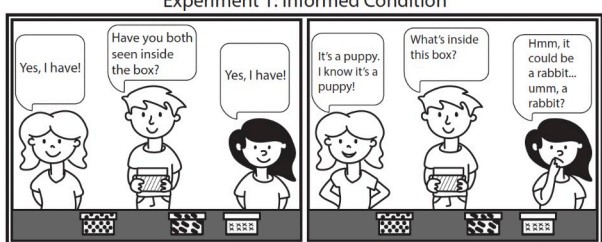

Experiment 1: Uninformed Condition

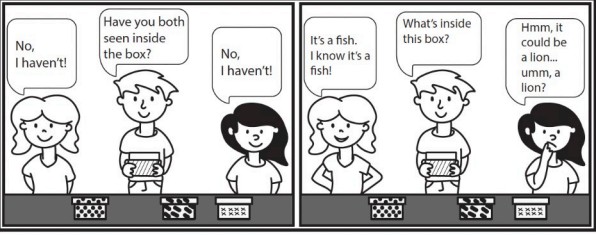

History Phase (Above)

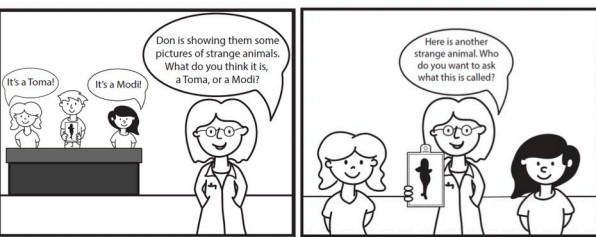

Test Phases: Endorse (Above Left) and Ask (Above Right)

**Fig 2. Visual schematic of Experiment 1 method.**

**Manipulation check.**   Our critical manipulation was based on children's understanding of the 'looking leads to knowing' principle–that is, someone one who has seen inside a box will know what is inside, whereas someone who has not seen inside will not know. Thus, we first examined children's responses to the post-test questions that served as manipulation checks. We found that the majority of participants (90.6%) correctly recalled whether or not the models had seen inside the boxes during the History Phase. The majority of children also correctly recalled whether the model had been confident (84%) or hesitant (82%) during the History Phase. Below we report all analyses with the full sample of participants, without exclusions. The same pattern of results emerges when excluding participants who failed the manipulation check (see S1 File).

**Who do children prefer to learn from?.**   Children's responses (0 = Hesitant model; 1 = Confident model) to the four *Ask* and four *Endorse* trials were modeled simultaneously with a random-intercept logistic regression with participant ID as a random effect to account for the repeated responses using *lme4* [42] in R [43] (Table 2). In Model 1, we find that children's learning preferences vary by condition: In the Informed Condition the odds that children preferred to learn from the previously Confident model was 21% greater than from the previously Hesitant Model. This preference to learn from the more confident model is consistent with the results of earlier work showing that children, like adults, capitalize on the 'confidence heuristic' and expands on this earlier work in a number of important ways (refer to the 'General Discussion'). In comparison, in the Uninformed Condition (*OR* = 0.83, .95CI = [0.72, 0.94], *p* = .004) the predicted odds of learning from the previously Confident model decreased by 17% compared to the Informed Condition.

**Table 2. Regression analyses on children's learning preferences in Experiment 1.**

| | Model 1 | | Model 2 | |
|---|---|---|---|---|
| | OR (.95%CI) | p | OR (.95%CI) | p |
| Intercept | 1.21 (1.08–1.35) | .001 | 1.04 (0.91–1.19) | .531 |
| Condition (1 = Uninformed) | 0.83 (0.72–0.94) | **.004** | 0.83 (0.73–0.95) | **.006** |
| Trial Type (1 = Endorse) | 1.01 (0.89–1.14) | .934 | 1.01 (0.89–1.14) | .921 |
| Age (Years, Scaled) | | | 1.04 (0.98–1.11) | .218 |
| Andrea Confident (1 = Yes) | | | 1.31 (1.15–1.49) | < .001 |
| $N_{ID}$ | 501 | | 501 | |
| Observations | 3922 | | 3922 | |

Importantly, the effect of condition was robust to the addition of control variables (Model 2). Even though both models were confident when making their statements during the Endorse Phase, children had a slight preference to favor the previously Hesitant model in the Uninformed Condition, suggesting they recognized that she was the better calibrated model. Finally, no clear developmental pattern emerged in children's learning preferences in the regression model. Post-hoc tests treating age categorically revealed that three- and four-year-olds did not differ in their tendency to favor the confident model between conditions: (Informed Condition: $M = .53$, $SD = .21$ versus Uninformed Condition: $M = .51$, $SD = .22$), $t$ (160) = -.725), $p = .469$, $ns$. In contrast, both the 5- and 6-year-old group, and those age 7 and older, were significantly less likely to choose the confident informant in the Uninformed condition (5–6 year olds $M = .48$, $SD = .18$; Ages 7+: $M = .51$, $SD = .12$) than in the Informed Condition (5–6 year olds $M = .57$, $SD = .19$, 7+: $M = .55$, $SD = .18$), $t$ (146) = -2.744, $p = .004$ and $t$ (189) = -1.786, $p = .038$, directional, respectively.

**Who do children think is smarter?.** Finally, we tested whether children differed between conditions in trait judgments of which model they thought was smarter (forced-choice). If children took into account a model's calibration, we would expect children to judge the Confident model as smarter in the Informed Condition and the Hesitant model as smarter in the Uninformed Condition. We also tested whether their choice of who was smarter varied with age. We found that the odds of the confident model being judged as 'smarter' were 2.51 times greater (.95CI = [1.90, 3.35], $p < .001$) in the Informed Condition (see Table 3, Model 1).

In comparison, the odds that children thought the Confident model was smarter in the Uninformed condition were 54% less (.95CI = [0.31, 0.66], $p < .001$). This effect was robust to the addition of controls (Model 2). Importantly, this effect of condition was significantly moderated by age (Model 3). In the Uninformed Condition, the odds of judging the Confident

**Table 3. Regression analyses on children's smartness judgments in Experiment 1.**

| Outcome (1 = Confident Model) | Model 1 | | Model 2 | | Model 3 | |
|---|---|---|---|---|---|---|
| | OR (.95%CI) | p | OR (.95%CI) | p | OR (.95%CI) | p |
| Intercept | 2.51 (1.90–3.35) | < .001 | 1.74 (1.24–2.44) | **.001** | 1.73 (1.24–2.45) | **.001** |
| Condition (1 = Uninformed) | 0.46 (0.31–0.66) | < .001 | 0.46 (0.31–0.67) | < .001 | 0.45 (0.31–0.67) | < .001 |
| Andrea Confident (1 = Yes) | | | 2.09 (1.43–3.06) | < .001 | 2.12 (1.45–3.13) | < .001 |
| Age (Years, scaled) | | | 0.92 (0.76–1.12) | .405 | 1.22 (0.92–1.66) | .175 |
| Condition * Age | | | | | 0.59 (0.40–0.88) | **.010** |
| Observations | 484 | | 484 | | 484 | |
| AIC | 628.95 | | 617.74 | | 612.83 | |

model as 'smarter' significantly decreased with age ($OR = 0.59$, $.95CI = [0.40, 0.88]$, $p = .01$). From Fig 3, we can estimate that children begin to significantly diverge in their judgments of which model is smarter based on calibration as of 4.8 years old, consistent with the developmental shift on learning preferences observed when treating age categorically. Five-year-olds in the Uninformed Condition are significantly less likely to judge the Confident (miscalibrated) model as smarter compared to those in the Informed Condition, and with age children increasingly judged the Hesitant (well-calibrated) model as 'smarter'.

## Discussion

The results from Experiment 1 suggest that by around 5 years of age children recognize that a person's history of being well-calibrated or miscalibrated bears on that person's credibility. This developmental finding is clearest in children's responses to the 'Who is Smarter?' question. Children's learning preferences also varied significantly by condition. That is, they were less likely to favor the previously Confident model in the Uninformed Condition (when she is miscalibrated) compared to the Informed Condition (when she is well-calibrated). Taken together, these results suggest that by around age 5 children can use calibration information to gauge a person's credibility.

One limitation in interpreting the results of Experiment 1 is that because children were required to choose between two informants (one previously confident and one previously hesitant) it is impossible to determine whether these results were driven by a tendency to *disfavor* the overly-confident informant or by a tendency to *favor* the justifiably hesitant model. A

**Fig 3. Probability of judging the confident model as 'smarter' in Experiment 1.** Shaded areas indicated confidence intervals.

similar limitation in interpretation emerged in a related study published while the current research was under investigation [44]. Kominsky, Langthorne, and Keil [44] were interested in examining when children understand the difference between what they referred to as 'mere ignorance' (i.e., not knowing something that is, in principle, knowable) and 'virtuous ignorance' (i.e., not knowing something because the knowledge is impossible or implausible to obtain). In their design, 5- to 10-year-olds were presented with silhouetted images of two informants along with descriptions of their answers to either 'knowable' (e.g., 'How many windows are in the White House?') or 'unknowable' questions (e.g. 'If you count all the leaves on all trees in the entire world, how many will you get?'). For each question, one respondent gave a specific answer (e.g., 'There are exactly 809,343,573,353,235 leaves on all trees in the world.'). In contrast, the other informant always admitted ignorance saying, "I don't know because it is not possible to answer that question precisely." For each question, participants were asked, "Which one do you think is the better expert?"

Kominsky et al. [44] found that only participants older than 9 years of age selected the ignorant informant for unknowable information, despite the fact that even the youngest children in their sample appeared to distinguish between knowable and unknowable items when asked how difficult the information would be to acquire. The authors suggest that by around age 9 children can distinguish between mere ignorance and virtuous ignorance. It is tempting to conclude from the findings of Kominsky et al. that children nine years of age and older appreciate that ignorance does not always indicate a lack of credibility. That is, one interpretation is that by around age 9 children understand that some people are *justifiably* ignorant (or to use their language, virtuously ignorant). However, an equally viable interpretation given the nature of Kominsky et al.'s design, like that used in Experiment 1 of the current paper, is that children do not understand anything about the ignorant informant but are simply avoiding the overly confident informant. Put differently, perhaps the 9-year-olds are not recognizing that admitting ignorance to an unknowable question is correct (or virtuous) but instead are wary of trusting informants who offer up impossible claims.

It is also possible that children in Experiment 1 might have a *better*, or earlier, understanding of justified hesitancy than their learning preferences suggest, but find it difficult to overcome a more general bias to avoid information from hesitant individuals. Even adults are biased to avoid hesitant individuals [e.g., 21–23] and in our design children would have to overcome any hesitancy avoidance to favor the well-calibrated model. To address these limitations in Experiment 1, Experiments 2 and 3 equated the models' level of confidence during the History Phase (both confident or both hesitant) and varied which of the models was informed. In Experiment 2, both models were *equally confident* throughout the experiment and we manipulated which of the two was informed (through visual access) during the History Phase. In other words, during the History Phase one model was Confident and Informed (well-calibrated) and the other was Confident and Uninformed (miscalibrated).

## Experiment 2

Experiment 2 investigated whether children would demonstrate an appreciation for calibration when deciding between two confident individuals: One whose confidence was justified (well-calibrated) and the other whose confidence was unjustified (miscalibrated). By comparing two confident individuals (rather than a confident versus a hesitant individual as in Experiment 1), we sought to overcome potential limitations in interpreting the results of Experiment 1. We focused Experiment 2 on a smaller sample of children narrowing our age range to include only ages 4–8, given that the regression analyses in Experiment 1 revealed no overall developmental differences and analyses treating age categorically showed that any

developmental changes in children's sensitivity to unjustified confidence were occurring between the youngest and the next youngest age groups (Mean age = 3.93 vs. 5.97 years, respectively).

## Method

**Participants.** Participants included 84 children (55% female) from 4 through 8 years ($M$ = 6.01 years, $SD$ = 1.55, range = 4.08–8.83) recruited through the Psychology Department's child participant pool ($n$ = 63) or the community ($n$ = 21, e.g., museum or local daycare). Design variables (e.g., model identity, speaking order, novel words set order) were counterbalanced within three age groups: Thirty-three 4-year-olds ($M$ = 4.49 years, $SD$ = .27; 61% female), 23 5- and 6-year-olds ($M$ = 5.97 months, $SD$ = .50; 57% female), and 28 7- and 8-year-olds ($M$ = 7.84 years, $SD$ = .62; 46% female). Eighty-eight percent of participants' parents reported their child's ethnicity: Caucasian (53.7%), Asian (28.9%), and mixed ethnicities (28.4%). An additional 4 children were tested but their data excluded because of language problems ($n$ = 1) or because they did not understand or complete the task ($n$ = 3). One child did not answer the 'Who is Smarter?' question. The study procedure was approved by the Behavioral Research Ethics Board in the Office of Research Services at University of British Columbia. Parents of participating children provided written consent and children provided verbal assent to participate. These data were collected between June 2015 to December 2016.

**Materials and procedure.** The procedure was identical to that in Experiment 1 except for the following critical differences: (1) The History Phase was altered to equate the models' level of confidence (i.e., both models were confident) and (2) instead we manipulated who saw the pictures inside the boxes as a within-subjects variable. During the 'History' Phase, participants watched a video of the two models being presented with four covered boxes by a male actor. Critically, one model was shown the boxes' contents and the other model was not shown the contents. For each box, the male actor asked the models (a) whether or not they had seen inside the box ("Have you seen inside this box?") to which the models correctly responded "yes" or "no"; and (b) to name what animal picture was inside the box ("What's inside this box?"). For each box, the models provided conflicting answers regarding the boxes' contents (e.g., rabbit vs. puppy). Importantly, both models responded confidently when asked to state the boxes' contents by providing non-verbal (e.g., head nodding), paralinguistic (e.g., declarative tone), and verbal cues of confidence (e.g., "It's a rabbit. I know it's a rabbit. It's a rabbit for sure!"). Unlike in Experiment 1, we did not indicate whether or not the models were correct when stating the boxes' contents, because both models were confident during the History Phase. In Experiment 2 children needed to infer whether the models' answers were correct from the models' visual access alone. Consequently, we could also ask participants to choose between the models' answers during the History Phase ("What do you think it's a picture of? A rabbit or a puppy?") using the same wording as used in the Endorse trials, resulting in three types of test trials: History, Endorse, and Ask trials. The History trials served as a manipulation check of whether one model was more knowledgeable than the other about the *current* contents of the boxes. In contrast, the Endorse and Ask trials tested whether children used the models' history of confidence calibration when deciding from whom to learn *new information*. Additionally, because both models were equally confident during the History Phase, the rationale in Experiment 1 for having the Endorse trials precede the Ask trials (i.e., to demonstrate the previously hesitant model was only hesitant when she lacked knowledge) no longer applied. Thus, we counterbalanced the Ask and Endorse trials in the Test Phase for Experiment 2. As in Experiment 1, post-test questions assessed recall of which model had seen inside the box ("Did Andrea [Beverly] see inside the boxes?") and which model children recognized

## Experiment 2: Informed or Uninformed Both Confident

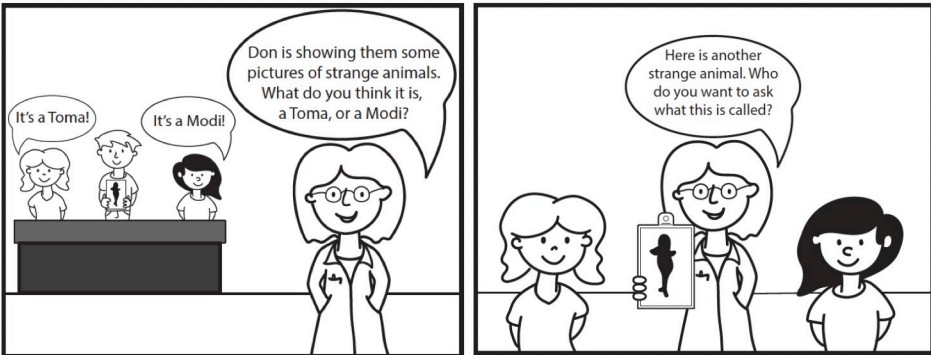

**History Phase (Above)**

**Test Phases: Endorse (Above Left) and Ask (Above Right)**

**Fig 4. Visual schematic of Experiment 2 method.**

as more credible ("Who is smarter? Andrea or Beverly?"). See Fig 4 for a visual schematic of the methods.

## Results

**Manipulation check.** Again, our critical manipulation was based on children's understanding of the 'looking leads to knowing' principle. We found that the majority of participants (74%) correctly recalled whether or not the models had seen inside the boxes during the History Phase (post-test items 1 and 2). Below we report all analyses with the full sample of participants, without exclusions. The same pattern of results emerges when excluding participants who failed the manipulation check (see S2 Table and S3 Table).

**Who do children prefer to learn from?.** The proportion of trials out of four for the History, Endorse, and Ask trials served as our learning dependent measures. Proportions were used to allow for the occasional trial in which a child did not respond or said "I don't know". Responses were coded such that higher scores in the learning trials indicated a preference to learn from the well-calibrated (justifiably confident) model (i.e., the one with visual access). Preliminary analyses ruled out effects of trial order, speaking order, novel word set order, location, or participant sex (all $ps > .19$), thus subsequent analyses collapsed across these variables.

Children's responses (0 = Model with No Visual Access; 1 = Model with Visual Access) to the four trials in each of the three phases, *History, Endorse, and Ask*, were modeled simultaneously with a random-intercept logistic regression with participant ID as a random effect to

**Table 4. Regression analyses on children's learning preferences in Experiment 2 History Phase.**

| | Model 1 | | | Model 2 | | |
|---|---|---|---|---|---|---|
| Predictors | Odds Ratios | CI | p | Odds Ratios | CI | p |
| (Intercept) | 1.43 | 1.10–1.86 | **0.007** | 1.38 | 0.97–1.97 | 0.070 |
| Age (years, scaled) | | | | 1.14 | 0.88–1.48 | 0.324 |
| Model Identity (1 = Andrea Knows) | | | | 1.07 | 0.64–1.80 | 0.788 |
| Observations | 335 | | | 335 | | |
| N | 84 | | | 84 | | |

account for the repeated responses using *lme4* [42] in R [43]. In the History phase, children wisely preferred to learn from the model with visual access (OR = 1.43 [1.10, 1.86]; see Table 4, Model 1). This effect was somewhat reduced when controlling for age and model identity (see Table 4, Model 2); but stronger when excluding children who failed the manipulation check (see S2 Table). In all models, we find some evidence of increases with age in preferences for the model with visual access (ORs > 1), but age was not a significant predictor.

In the Ask and Endorse trials there was also a clear preference for learning from the justifiably confident model (OR = 1.34 [1.07, 1.68]). We did not find that children's preferences differed between the ask and endorse trials. The preference for learning from the justifiably confident model was robust to the addition of covariates for age and model identity (see Table 5, Model 2), and to manipulation check exclusions (see S3 Table).

Again, age was not a significant predictor of children's performance. Additional analyses treating age categorically with the 3 age groups (i.e., 4-year-olds, 5-6-year-olds, and 7-8-year-olds) confirmed that there was no main effect of age ($p = .705$) and no interaction between age and trial type (History, Endorse, Ask; $p = .509$). Overall, children preferred to learn from the justifiably confident model at above chance levels across all three types of learning trials: History Trials ($M = .58$, $SD = .28$, $t (83) = 2.670$, $p = .009$), Ask Trials ($M = .58$, $SD = .18$, $t (83) = 3.864$, $p < .001$), and Endorse Trials ($M = .58$, $SD = .26$, $t (83) = 2.824$, $p = .006$). Even the 4-year-olds' mean learning across all future learning trials (Ask and Endorse) was greater than chance, $t (32) = 1.966$, $p = .029$, directional. Therefore, by at least age 4 children appreciate that someone whose confidence is well-calibrated is a more trust-worthy source of new information than someone whose confidence is miscalibrated. It remains an open question for future research to identify precisely when in development children *first* use whether one's confidence is well-calibrated to guide their learning.

**Who do children think is smarter?.** Finally, we tested whether children's trait judgments of which model they thought was smarter (forced-choice) took into account the models' calibration. If so, we would expect children to judge the model whose confidence was previously justified (because she was visually informed) as smarter than the model whose confidence was

**Table 5. Regression analyses on children's learning preferences in Experiment 2 ask and endorse trials.**

| | Model 1 | | | Model 2 | | |
|---|---|---|---|---|---|---|
| Predictors | Odds Ratios | CI | p | Odds Ratios | CI | p |
| (Intercept) | 1.34 | 1.07–1.68 | **0.010** | 1.52 | 1.17–1.99 | **0.002** |
| Trial (1 = Endorse) | 1.03 | 0.76–1.41 | 0.834 | 1.03 | 0.76–1.41 | 0.831 |
| Age (years, scaled) | | | | 1.06 | 0.91–1.24 | 0.460 |
| Model Identity (1 = Andrea Knows) | | | | 0.76 | 0.55–1.04 | 0.091 |
| Observations | 668 | | | 668 | | |
| N | 84 | | | 84 | | |

**Table 6. Regression analyses on children's smartness judgments in Experiment 2.**

| Predictors | Model 1 | | | Model 2 | | |
|---|---|---|---|---|---|---|
| | Odds Ratios | CI | p | Odds Ratios | CI | p |
| (Intercept) | 2.77 | 1.70–4.51 | <**0.001** | 5.35 | 2.38–12.06 | <**0.001** |
| Age (years, scaled) | | | | 1.12 | 0.67–1.87 | 0.678 |
| Model Identity (1 = Andrea Knows | | | | 0.30 | 0.10–0.84 | **0.022** |
| N | 83 | | | 83 | | |

previously unjustified (because she was not visually informed). We also tested whether their choice of who was smarter varied with age. Preliminary analyses ruled out any effects of trial order, speaking order, novel word set order, location, or participant sex (all *ps* >.23).

We found that the majority of children (73%) indicated that the justifiably confident model was 'smarter'. After controlling for age and model identity, the odds of the well-calibrated model being judged as 'smarter' were 5.35 times greater (.95CI = 2.38, 12.06], *p* < .001) than the odds that the miscalibrated model was judged as smarter (see Table 6, Model 2). After exclusions for failing the manipulation checks, all but 8 children thought that the justifiably confident model was smarter.

Age did not significantly predict children's choice of who was smarter. Additional analyses treating age categorically as above revealed no significant differences by age, *F* (2, 80) = 1.534, *p* = .222. Even the 4-year-olds chose the justifiably confident model as smarter 67% of the time (*M* = .67, *SD* = .48); more often than chance, *t* (32) = 2.000, *p* = .027, directional.

## Discussion

The results from Experiment 2 indicate that children ages 4–8 years understand that confidence should be calibrated with one's knowledge. By directly comparing two confident models (rather than a confident and hesitant model as in Experiment 1), we found that children as young as 4 years of age used a person's history of calibration as a cue to a model's credibility and preferred to learn from a well-calibrated confident model over a miscalibrated (overconfident) model. In other words, when deciding between justifiably and unjustifiably confident individuals, children as young as 4 years prefer the individual whose confidence is justified. Considering the results of Experiments 1 and 2 together, we found that 4-year-olds preferred the well-calibrated confident model in Experiment 2, yet similarly aged children did not show a preference for the well-calibrated hesitant model in Experiment 1, which suggests that young children may be especially hesitancy-avoidant, or struggle to understand calibration as it applies to hesitancy.

To test this interpretation, we investigated whether children's sensitivity to calibration extends to *hesitant* models. Experiment 2 demonstrated that children as young as age 4 appreciate that to be credible a model's confidence must be *justified*! Do children appreciate that a model who is hesitant can also be credible if that hesitancy is justified? It seems plausible that children do not acquire a singular concept of calibration but instead have two separate concepts that may have different developmental onsets: 1) overly confident people are not credible; a person's confidence must be justified, and 2) hesitant people can also sometimes be credible, if their hesitancy is justified.

## Experiment 3

Experiment 3 investigated whether children ages 5–8 would demonstrate an appreciation for calibration when deciding between two hesitant individuals—one whose hesitance was justified (well-calibrated, because she did not see inside the boxes) and the other whose hesitance was

unjustified (miscalibrated, because she did see inside the boxes). In the absence of prior research examining one's sensitivity to calibration in *hesitant* models we did not have an a priori prediction of precisely when in development children would favor a well-calibrated hesitant model but we suspected that it would be a later-developing accomplishment. We therefore started with a sample of children ages 5–8 and analyzed their data before deciding whether there was any value in testing a younger group.

## Method

**Participants.** Participants included 76 children (54% female) 5 through 8 years of age ($M$ = 7.11 years, $SD$ = 1.16, range = 5.0–8.92) recruited through the Psychology Department's child participant pool ($n$ = 42) or the community ($n$ = 34; e.g., museum or daycare). Forty-four 7- and 8-year-olds ($M$ = 7.97 years, $SD$ = .57, 59% female) and 32 5- and 6-year-olds ($M$ = 71.13 months, $SD$ = 6.84, 47% female). Based on the analyses of the data with 5–8 year olds (i.e., to foreshadow, neither age group was sensitive to hesitancy calibration) there was no reason to include younger participants in our sample. Eighty-percent of participants' parents reported their child's ethnicity: Caucasian (39.3%), Asian (41%), and mixed ethnicities (19.7%). An additional 4 children were tested but their data excluded because of experimenter error ($n$ = 1) or because they did not understand or complete the task ($n$ = 3). One child did not answer the 'Who is Smarter?' question. The study procedure was approved by the Behavioral Research Ethics Board in the Office of Research Services at University of British Columbia. Parents of participating children provided written consent and children provided verbal assent to participate. These data were collected between March and September 2017.

**Materials and procedure.** The procedure was identical to that in Experiment 2, except that both models were *equally hesitant* in the History Phase. That is, during the History Phase, although one model saw inside the four boxes and the other did not, both models responded hesitantly when asked to state the boxes' contents by providing non-verbal (e.g., head tilting), paralinguistic (e.g., using a slow rate of speech with pauses), and verbal cues of hesitancy (e.g., "Umm, It could be a puppy. . ..hmm. . .maybe a puppy? I'll guess a puppy?"). Note that because the remainder of the experiment was the same as Experiment 2, both previously hesitant models were equally confident when offering their answers in the Endorse Phase (when they could see the animal picture), providing evidence that these models are not always hesitant, and requiring that any bias to favor learning from one model stems from the model's *history* of calibration.

See Fig 5 for a visual schematic of Experiment 3 method.

## Results

**Manipulation check.** The majority of children correctly recalled whether or not the models had seen inside the boxes (87% correctly answered both post test questions 1 and 2). Again, we report all analyses with the full sample of participants, without exclusions, below. The same pattern of results emerges when excluding participants who failed the manipulation check (see S4 Table, S5 Table, and S6 Table).

**Who do children prefer to learn from?.** The proportion of History, Endorse, and Ask trials in which the participant chose the model with visual access again served as our dependent measures of learning preferences. Responses were coded such that higher scores indicated a preference to learn from the informed (unjustifiably hesitant) model. Preliminary analyses revealed no effects of sex, trial order, location, speaking order, or which word set came first ($p$s > .08), therefore, we collapsed across these variables. For History trials participants were expected to endorse the visually-informed model (the one who saw inside), whereas the

## Experiment 3: Informed Or Uninformed Both Hesitant

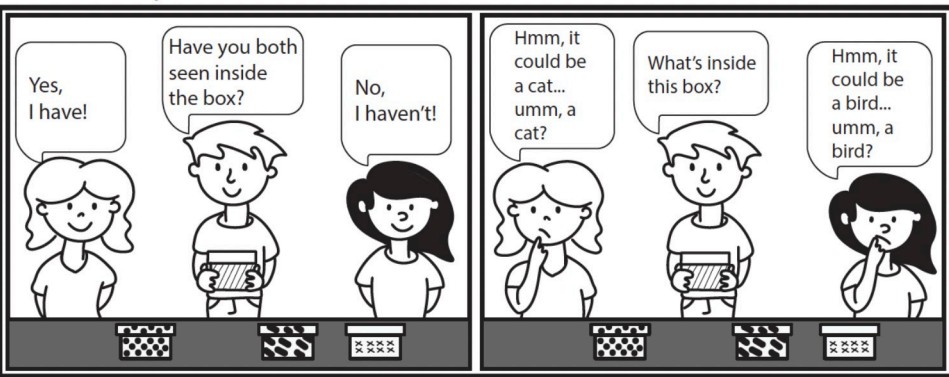

History Phase (Above)

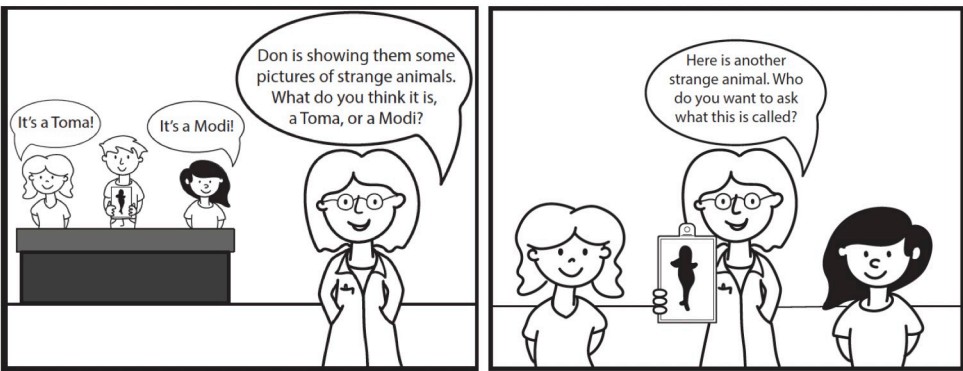

Test Phases: Endorse (Above Left) and Ask (Above Right)

**Fig 5. A visual schematic of Experiment 3 method.**

opposite pattern should emerge for the Endorse and Ask Trials if children were sensitive to well-calibrated versus miscalibrated hesitancy.

In the History phase, children preferentially learned from the informed model (i.e., the one with visual access; OR = 1.32 [1.04, 1.67]; see Table 7, Model 1). In Model 2, we find this preference increased with age (OR = 1.34 [1.05, 1.70]; Table 7). This developmental effect was slightly weaker after manipulation check exclusions, but the overall preference for the informed model was robust to these exclusions (see S4 Table).

In the Endorse and Ask trials, however, we find no clear evidence for a preference for either the previously informed or uninformed model (see Table 8 and S5 Table). Additional analyses treating age categorically using the 2 age groups (i.e., 5–6 years and 7–8 years) confirmed no

**Table 7. Regression analyses on children's learning preferences in Experiment 3 History Phase.**

|  | Model 1 | | | Model 2 | | |
|---|---|---|---|---|---|---|
| *Predictors* | *Odds Ratios* | *CI* | *p* | *Odds Ratios* | *CI* | *p* |
| (Intercept) | 1.32 | 1.04–1.67 | **0.021** | 1.45 | 1.02–2.07 | **0.040** |
| Age (years, scaled) |  |  |  | 1.34 | 1.05–1.70 | **0.018** |
| Model Identity (1 = Andrea Knows) |  |  |  | 0.85 | 0.53–1.37 | 0.502 |
| Observations | 283 | | | 283 | | |
| N | 72 | | | 72 | | |

**Table 8. Regression analyses on children's learning preferences in Experiment 3 ask and endorse trials.**

| Predictors | Model 1 | | | Model 2 | | |
|---|---|---|---|---|---|---|
| | Odds Ratios | CI | p | Odds Ratios | CI | p |
| (Intercept) | 1.12 | 0.89–1.41 | 0.348 | 1.06 | 0.79–1.42 | 0.681 |
| Trial (1 = Endorse) | 0.96 | 0.69–1.33 | 0.808 | 0.96 | 0.69–1.33 | 0.813 |
| Age (years, scaled) | | | | 1.10 | 0.93–1.30 | 0.260 |
| Model Identity (1 = Andrea Knows) | | | | 1.10 | 0.79–1.53 | 0.586 |
| Observations | 576 | | | 576 | | |
| N | 73 | | | 73 | | |

effect of age group (*p* = .410), and no interaction between age and trial type (*p* = .882). That is, regardless of age, children did not show a preference to favor the better calibrated hesitant model on either the Ask (*M* = .47, *SD* = .17) or the Endorse trials (*M* = .48, *SD* = .19).

**Who do children think is smarter?.** Next, we tested whether children differed in trait judgments of which model they thought was smarter (coded as 1 = miscalibrated model and 0 = well-calibrated model). Preliminary analyses revealed no effects of sex, trial order, location, speaking order, or which word set came first (ps > .27), therefore we collapsed across these variables.

We found that children did not consistently think the informed or uninformed model was smarter. The only significant predictor was model identity, favoring Andrea as the smarter model over Beverly (see Tables 9 and S6). Additional analyses treating age categorically, confirmed that neither the 7- to 8-year-olds (*M* = .41, *SD* = .50) nor the 5- to 6-year-olds (*M* = .42, *SD* = .50) showed a significant preference for the well-calibrated hesitant model when selecting who they thought was smarter, *t* (43) = -.895, *p* = .378 and *t* (30) = -1.212, *p* = .232, respectively.

**Analyses comparing Experiments 2 and 3.** The test phases of Experiments 2 and 3 were identical. The two experiments differed only in the manipulation applied during the History Phase allowing us to directly compare children's performance in Experiments 2 versus 3. These analyses provide a direct test of whether discriminating a well-calibrated model from a miscalibrated model is harder when the two models are hesitant (Experiment 3) than when the two models are confident (Experiment 2). Using logistic regression we predicted children's preferences to learn from the well-calibrated model on the Ask and Endorse trials. Clear preferences for the well-calibrated model in Experiment 2 were observed when the two models were confident (Model 1 Intercept: *OR* = 1.36, .95CI = [1.12, 1.64], *p* = .002), but singling out the well-calibrated model when both were hesitant in Experiment 3 appears considerably more difficult (Model 1 Study: *OR* = 0.64, .95CI = [0.52, 0.80], *p* < .001; see Table A in S2 File).

Similarly, logistic regression models predicting who children thought was smarter in Experiments 2 and 3 found that the odds of attributing the calibrated model with intelligence were

**Table 9. Regression analyses on children's smartness judgments in Experiment 3.**

| Predictors | Model 1 | | | Model 2 | | |
|---|---|---|---|---|---|---|
| | Odds Ratios | CI | p | Odds Ratios | CI | p |
| (Intercept) | 1.52 | 0.95–2.42 | 0.081 | 0.80 | 0.41–1.59 | 0.529 |
| Age (years, scaled) | | | | 1.18 | 0.72–1.94 | 0.512 |
| Model Identity (1 = Andrea Knows) | | | | 3.57 | 1.32–9.63 | **0.012** |
| N | 73 | | | 73 | | |

vastly greater in Experiment 2 where both models were confident. Full details of the analyses comparing Experiments 2 and 3 is provided in S2 File.

## Discussion

The results of Experiment 3 suggest that during the History trials the older children wisely paid attention to which of the two models had visual access to the contents of the boxes and chose to trust the statements about the boxes contents from the one who had seen inside. During the History Phase the models' calibration is irrelevant and visual access is the cue that is most informative as to which model is the most knowledgeable about the boxes' contents. Interestingly, children showed no preference, in either direction, on the Ask and Endorse trials. Even the oldest children (7- to 8-year-olds) did not shown any evidence of using the models' history of hesitancy calibration when learning new information. If they were sensitive to hesitancy calibration they should have favored the previously well-calibrated model (i.e., the one who was only hesitant when she had *not* looked inside) rather than the one who had previously expressed uncertainty even when looking inside. In sum, the results of Experiment 3 and the comparative analyses between Experiments 2 and 3 suggest that until at least age 8 children are insensitive to calibration when interpreting a model's hesitancy and that hesitancy calibration is a much later developing understanding than children's sensitivity to confidence calibration, which appears to be in place by around age 4.

## General discussion

The primary goal of the research outlined in this manuscript was to examine how children's learning decisions and credibility judgments are affected by an individual's calibration, where calibration is an index of the relationship between a) the level of confidence (or hesitancy) expressed by an individual and b) whether that individual is knowledgeable or not about the information they provided. This research addressed several questions: Do children appreciate that confidence is a sign of credibility *only if* that confidence is justified? And conversely, that hesitancy is *not* indicative of a lack of credibility if that hesitancy is justified? Moreover, how does an individual's calibration influence children's learning decisions and impressions of their intelligence? This research adds to growing bodies of research on children's trait judgments and impression formation, children's understanding of information access cues (namely visual access as a cue to knowledge), and children's selective social learning preferences. As such, this research contributes to a more comprehensive understanding of children's social cognition and how it changes with development.

Three findings of interest emerged in Experiment 1. First, given a choice to learn from a previously Confident model over a previously Hesitant model, children in the Informed Condition generally preferred to learn from a previously Confident model. This is consistent with a wealth of research on the 'confidence heuristic' in adults and adds to the small but growing body of work with young children [e.g., 31–33, 36]. These results expand on earlier work with children by using a design that examines both children's learning preferences and their explicit judgements of the models' smartness. It also expands on earlier work on children's sensitivity to confidence by using a larger sample with a broader range of ages. Second, we provide evidence that children's learning decisions can be influenced by a model's calibration: Children were less likely to learn from a Confident model in the Uninformed Condition (when her confidence was not justified; she was miscalibrated) than they were to learn from the Confident model in the Informed Condition. Third, children's impressions of an individual's smartness were also influenced by a model's calibration: By around age 5 children were significantly less likely to judge the Confident model as smarter when she was miscalibrated (Uninformed

Condition) compared to when she was well-calibrated (Informed Condition). Moreover, with age children increasingly judged the Hesitant (well-calibrated) model as smarter.

Experiment 1's finding that preschoolers can differentiate between informants whose confidence is, or is not, justified, is also consistent with new work published since the present research was completed [44]. Huh et al. [45] pitted a cautious speaker's statement against a confident speaker's statement and preschool children were given a choice of whose opinion to trust about the best shop for ice-cream. When both speakers were informed children trusted the confident speaker's opinion, however when both were only partially informed (rendering the person's confidence less justified) children did not believe the confidence person over the cautious one; they showed no preference for either informant. Importantly, the current research differs from, and extends, the work by Huh et al. [45] by examining children's learning of generic information (i.e. category labels, e.g., labels for novel animals) rather than opinion-based information (i.e., which shop has the best ice-cream). The current research also examined how a speaker's calibration influences children's impressions of their intelligence (i.e., who is smarter?) and, critically, goes beyond pitting a confident individual against a hesitant one, to examine children's understanding of calibration separately for confidence versus hesitancy.

Specifically, the results from Experiments 2 and 3 disentangled whether children's sensitivity to calibration in Experiment 1, and Huh et al [45], was driven by a bias to avoid an overly-confident informant or to favor a justifiably hesitant one by revealing a clear bias to avoid an overconfident informant but no evidence of favoring a justifiably hesitant one. Experiment 2 demonstrated that 4- to 8-year-old children are sensitive to whether a model's confidence is well-calibrated and they are wary of learning from an overly confident model. The confident model's calibration also influenced children's judgments of the model's smartness. In contrast, even 7- & 8-year-old children's learning decisions and smartness judgments were unaffected by a model's hesitancy calibration. Adults, such as yourself, understand that sometimes hesitancy is justified—this understanding was not evident even in the older children in our sample suggesting it is a later-developing understanding (i.e., it only appears at some point after age 8). Future research will benefit from identifying precisely when in development people first become sensitive to hesitancy calibration and how it affects their learning decisions and impressions.

We want to acknowledge the limits of drawing conclusions from null results in Experiment 3 and from significant differences between experiments. The differences between the results of Experiment 2 and 3 suggest a developmental difference in understanding confidence versus hesitancy calibration, but we recognize this may not be a perfect comparison. Although the designs are largely identical, perhaps being hesitant with visual access is *more* forgivable than being confident without visual access. The speaker might be looking at something that is unclear or unknown to her that could account for her hesitancy even in the informed condition. We find this alternative explanation unlikely since participants also did not prefer to learn from an informed hesitant model over an uninformed one *within* Experiment 3. Under the above 'forgiveness' interpretation, they would also be failing to judge that completely uninformed hesitancy is more forgivable than partially-informed hesitancy, or at least a 'safer bet'. Nonetheless, we urge caution in interpreting the results of Experiment 3 and call upon future research to investigate children's understanding of justified versus unjustified hesitancy using alternative methods.

Of note, the results of Experiments 1 and 2 raise the interesting question as to why our results of sensitivity to confidence calibration (Experiments 1 and 2) differ from Tenney et al. [29]. Recall that Tenney et al. found that 5- and 6-year-olds were insensitive to a witness's calibration and instead believed the witness who was consistently confident (but miscalibrated, in

this case *over*confident), over the witness who was less confident overall but better calibrated. We suspect that there are two aspects of Tenney et al.'s design that may have contributed to young children's apparent insensitivity to calibration. First, the difference in the calibration of the eyewitnesses was based on only a single statement (e.g., whether the incorrect statement about the color of the ball was said with confidence or with hesitancy). Although sometimes a single mistake is sufficient to influence children's selective learning decisions [46], more often forming trait impressions relies on several repeated instances [47]. In our design, we provided four repeated demonstrations of one model being well-calibrated and the other being miscalibrated. The second aspect of Tenney et al.'s design that may have made it difficult for young children was that the accuracy of the eyewitnesses' statements was unknown to participants at the time the witnesses made the statements. Participants first judged which witness they believed, then were told about the accuracy of the eyewitnesses' earlier statements and asked a second time which witness they believed. To detect the lack of calibration in the one witness's statement they would have to *retroactively* incorporate their new knowledge of the statement's accuracy with the witness's earlier level of confidence. Retroactive application of information about a model's credibility can be difficult for young children as it requires additional memory and cognitive demands [48]. In comparison, in our design the model's calibration could be inferred *at the time* of the models' initial statements, likely making it much easier to recognize the model's calibration.

Importantly though, the fact that these design differences can result in different findings regarding children's recognition of confidence calibration serves to highlight that it may be harder to detect calibration in some contexts. It is likely not as simple as saying that by four or five years of age children *will always* capitalize on confidence calibration information to make inferences about a person's credibility. Indeed, as Tenney et al. [29] observed, even adults are persuaded by (over)confidence when under cognitive load. That such differences exist even in adults serves as a reminder that when considering social cognitive functioning we not only have to consider age differences and individual differences but we should also bear in mind contextual and *inter-subject* differences (such as one's level of attention or motivation at any given point in time). In other words, there may be differences between what children *can do* (in theory) and what they *actually do* (in practice) in any given situation. The real world is more complex than a controlled lab setting, and therefore other variables may come into play. For instance, in our design the models' history of being well- or miscalibrated was demonstrated over a series of repeated trials in quick succession, whereas in many real-world sessions (excluding courtroom testimony and political debates, for instance) tracking a person's calibration may take place over a much longer period of time interrupted by numerous other cognitive demands. Additional factors to consider in one's use of calibration are a) the child's sensitivity to a person's prior accuracy and b) the child's memory abilities. A person's calibration only becomes evident by keeping track over time of one's history of the relationship between that person's prior accuracy and confidence. As such the child must possess the ability to detect accuracy and inaccuracy *and* detect confidence and hesitancy *and* detect and remember how the two were correlated. In such a way, in future instances when that person's accuracy is unknown, one can use that person's history of being well- or miscalibrated to *infer* their current or future accuracy. In our design, the test trials took place immediately after the history phase establishing the model's calibration. Real-world situations would likely involve remembering this information over longer periods of time. Moreover, in real world settings a person's level of calibration likely varies proportionally rather than being perfectly correlated. In our design both models showed a one-to-one relationship between their level of confidence and their underlying knowledge: that is, one model was *always* miscalibrated (always confident when uninformed, or hesitant when informed) and the other perfectly calibrated (always

confident when informed or hesitant when uninformed). In the real-world, the correlation is unlikely to be one-to-one. For example, Person A might be overconfident only 25% of the time.

It is also worth considering the implications of the model bias (favoring one of the adult models over the other) in our results. Could the absence of a calibration effect in Experiment 3 stem from the model bias overpowering its effect? Previous research has revealed that attractiveness influences children's selective learning preferences [49], and can even trump more direct cues of credibility such as a person's prior accuracy [50]. Of course, we cannot be certain the model bias in our research stemmed from attractiveness *per se*; it could have been about charisma, eye-gaze, vocal tone, affect, and so forth. Regardless of the cause of the model bias, it is interesting to note that it did not completely trump children's sensitivity to confidence calibration (Experiments 1 and 2). The same two models (see Fig 1) were used in all 3 experiments and which role they played (well-calibrated or miscalibrated) was always counterbalanced. Nonetheless a model bias, favoring Andrea, often occurred. When this model bias emerged in the learning measures it tended to be relatively small (e.g. In Experiment 1 the odd ratio of 1.31 equates to a 56.75% probability of favoring Andrea), though the model bias was larger in the smarter judgments, suggesting a possible halo effect. The model bias was largest when children were unable to use another cue (i.e., hesitancy calibration in Experiment 3) to guide their judgments. Importantly, children's sensitivity to calibration was significant, above and beyond any tendency to favor Andrea, in both Experiments 1 and 2. We think it is unlikely that the model bias can fully explain children's insensitivity to hesitancy calibration in Experiment 3, given that a calibration effect does not emerge even after controlling for model bias, whereas it did in Experiments 1 and 2.

Taken together, the findings from the present research fill an important gap in the literature: Children appear sensitive to a model's calibration from fairly early in development (by around age 4 or 5) but the age at which they use calibration to guide their learning preferences and credibility judgments seems dependent on whether the models are confident or hesitant. This research shows some important limits on children's understanding of calibration. Psychologists defined the concept of 'calibration' as the relationship between a person's expressed level of confidence and that person's underlying knowledge. Embedded within this concept seems to be the reasonable assumption that one's level of confidence is perceived on a continuum, with confidence at one end and hesitancy at the other end. Although this is a useful way to conceptualize this relationship, we posit that children do not develop a concept of calibration in this way (or do not fully understand confidence as a continuum). We propose that rather than understanding calibration as a singular concept, children may proceed through different stages of understanding.

Based on the results of the experiments outlined here, we cautiously suggest that a three-stage developmental progression occurs in which children come to understand the meaning of confidence versus hesitancy as two separate cues with different developmental onsets. Initially (in Stage 1) children only appear to recognize that a person's confidence is a useful cue to guide learning (i.e., confident people tend to be knowledgeable and reliable sources of information, whereas hesitant people do not). At this stage children will be heavily swayed by people who appear confident and fall prey to those who are overly confident. Later (in Stage 2), armed with sufficient evidence that confident people are not *always* good sources of information, they recognize that they should avoid confident sources when other cues suggest that their confidence is not reflective of their underlying knowledge. Moreover, we propose an asynchrony in children's understanding of calibration at this second stage, wherein children do not yet understand that hesitant people can sometimes be knowledgeable. That is, at this stage children will be less likely to learn from overly confident people *and* all hesitant people.

Finally, (at Stage 3) children come to recognize that hesitant people do not *always* lack knowledge. Heuristically these stages might work something like this: Stage 1: 'trust all confidence sources, ignore hesitant ones.' At Stage 2 this heuristic gets modified to 'trust *most* confidence sources; but watch out for overly-confident sources—those with a history of being confident when other evidence suggests they are not knowledgeable'. At Stage 3 there is a qualifier: 'don't ignore *all* hesitant sources, hesitant sources should not be ignored if their hesitancy is justified'.

Importantly, the two-stage model implied by Tenney and colleagues [29]–wherein children initially lack an appreciation of calibration and later, at least by adulthood, understand confident individuals can be miscalibrated–does not account for the developmental asynchrony in understanding calibration for confident versus hesitant individuals. We found evidence against this two-stage model, which implies the existence of *at least* one additional stage, which according to our work occurs after age 8. Whether or not a three-stage progression is an accurate depiction of how children conceive of the relations between an individual's credibility and their expressed level of confidence and hesitancy will require further investigation. The developmental changes noted in our research during the preschool period (representing a transition from stage 1 to 2) are consistent with the developmental change demonstrated by Brosseau-Liard et al. [36]. In their work, 4-year-olds were swayed by confident sources that were *previously confident when making inaccurate claims*, whereas by 5-years of age they recognized that accuracy was a better indication of a source's credibility, and were wary of learning new information from sources that were previously over-confident. Support for a developmental progression from Stage 2 to Stage 3 comes from the differences observed between Experiments 2 and 3 in the current research. To our knowledge no other research has specifically focused on children's understanding of justified hesitancy. Future research on how children reason about justified hesitancy would likely prove fruitful, in much the same way as research on how children reason about justified ignorance [3].

In conclusion, this research shows that by around four years of age children can distinguish between a model who is overly-confident and one who is justifiably confident and they prefer to learn from a justifiably confident person, and perceive her as smarter, than an overly-confident person. In comparison, even 7- and 8-year-old children seem unable to recognize a justifiably hesitant person as a credible source of information. In addition, this work shows how flexibly children navigated a variety of information sources to guide their learning decisions and deduce who was most credible. Specifically, the children in our design demonstrated the ability to integrate several cues that can license inferences about others' knowledge states, including 1) visual access cues, 2) the model's current confident states, and 3) the models' history of being calibrated, revealing a fairly sophisticated integration of social cognitive cues. Taken together this set of experiments provides a more comprehensive demonstration of children's complex and nuanced understanding of the mind while simultaneously illuminating specific limitations in their social cognitive understanding.

## Supporting information

**S1 Table. Experiment 1 participant demographic information by age years.**
(DOCX)

**S2 Table. Regression analyses on children's learning preferences in Experiment 2 History Phase with exclusions.**
(DOCX)

**S3 Table. Regression analyses on children's learning preferences in Experiment 2 ask and endorse trials with exclusions.**
(DOCX)

**S4 Table. Regression analyses on children's learning preferences in Experiment 3 History Phase with exclusions.**
(DOCX)

**S5 Table. Regression analyses on children's learning preferences in Experiment 3 ask and endorse trials with exclusions.**
(DOCX)

**S6 Table. Regression analyses on children's smartness judgments in Experiment 3 with exclusions.**
(DOCX)

**S1 File. Regression analyses on children's learning preferences in Experiment 1 excluding participants who failed the manipulation check.**
(DOCX)

**S2 File. Full details of the regression analyses comparing Experiments 2 and 3.**
(DOCX)

## Acknowledgments

We are grateful to Siba Ghrear, Taeh Haddock, Andrea Kim, Parky Lau, Dorna Rahimi, Alexandra Thomson, Keera Wanney, Megan Whyte, and Mimi Zhang for assisting in data collection. We thank Charlotte Stewardson for the artwork depicting the experimental procedures and Carolyn Baer for helpful feedback on an earlier draft of this manuscript.

## Author Contributions

**Conceptualization:** Susan A. J. Birch.

**Data curation:** Rachel L. Severson.

**Formal analysis:** Susan A. J. Birch, Rachel L. Severson, Adam Baimel.

**Funding acquisition:** Susan A. J. Birch.

**Investigation:** Susan A. J. Birch, Rachel L. Severson.

**Methodology:** Susan A. J. Birch, Rachel L. Severson.

**Project administration:** Susan A. J. Birch, Rachel L. Severson.

**Resources:** Susan A. J. Birch.

**Supervision:** Susan A. J. Birch, Rachel L. Severson.

**Validation:** Adam Baimel.

**Visualization:** Rachel L. Severson, Adam Baimel.

**Writing – original draft:** Susan A. J. Birch, Rachel L. Severson, Adam Baimel.

**Writing – review & editing:** Susan A. J. Birch, Rachel L. Severson.

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
