## [Decision Letter · Decision Letter 0]

3 Sep 2019

PONE-D-19-18550

Children’s understanding of when a person’s confidence and hesitancy is a cue to their credibility

PLOS ONE

Dear Dr. Severson,

Thank you for submitting your manuscript to PLOS ONE. After careful consideration, we feel that it has merit but does not fully meet PLOS ONE’s publication criteria as it currently stands. Therefore, we invite you to submit a revised version of the manuscript that addresses the points raised during the review process.

Please find below the reviewer's comments, as well as those from my own.

We would appreciate receiving your revised manuscript by Oct 18 2019 11:59PM. To enhance the reproducibility of your results, we recommend that if applicable you deposit your laboratory protocols in protocols.io, where a protocol can be assigned its own identifier (DOI) such that it can be cited independently in the future. For instructions see: http://journals.plos.org/plosone/s/submission-guidelines#loc-laboratory-protocols

We look forward to receiving your revised manuscript.

Kind regards,

Valerio Capraro

Academic Editor

PLOS ONE

Journal Requirements:

1. We note that you have stated that you will provide repository information for your data at acceptance. Should your manuscript be accepted for publication, we will hold it until you provide the relevant accession numbers or DOIs necessary to access your data. If you wish to make changes to your Data Availability statement, please describe these changes in your cover letter and we will update your Data Availability statement to reflect the information you provide.

2. We note that Figure [1] includes an image of a [patient / participant / in the study]. 

Additional Editor Comments (if provided):

I have now collected one review from one expert in the field. Unfortunately, I was unable to collect a second review. However, the one review I could collect is extremely thoughtful. Therefore, I am happy to make a decision with only one review. The reviewer asks for several revisions. Therefore, I would like to invite you to revise your paper following their comments. Needless to say that all comments must be addressed.

I am looking forward for the revision.

Reviewers' comments:

Reviewer's Responses to Questions

**Comments to the Author**

1. Is the manuscript technically sound, and do the data support the conclusions?

Reviewer #1: Partly

2. Has the statistical analysis been performed appropriately and rigorously? 

Reviewer #1: I Don't Know

3. Have the authors made all data underlying the findings in their manuscript fully available?

Reviewer #1: Yes

4. Is the manuscript presented in an intelligible fashion and written in standard English?

Reviewer #1: Yes

5. Review Comments to the Author

Reviewer #1: Below is a review of submission PONE-D-19-18550:

Children’s understanding of when a person’s confidence and hesitancy is a cue to their credibility

This paper reports on findings from three experiments looking at children’s understanding of calibration, the match between an informant’s confidence/hesitancy and knowledge. E1 asks whether 3 to 12-year-olds prefer to learn from “well-calibrated” informants (those who are confident when knowledgeable and hesitant when not), and whether children view well-calibrated informants as “smarter”. The remaining experiments follow-up E1 by equating the informant’s level of confidence (E2) and hesitancy (E3). Findings suggest that by age 4 or 5 children differentiate between well-calibrated from mis-calibrated informants (E1), provided those informants had a history of being confident (E2) rather than hesitant (E3). Overall the article is well-written, the findings are appropriate for the journal, and the topic is likely to be of interest to the journal’s readership. Below are several suggestions intended to help make future drafts of the paper stronger.

1) There are some inconsistencies across the 3 experiments, many of which are not clearly motivated or explained. For example:

a. The Endorse and Ask questions are counterbalance in E2 and E3, but not E1. Why not counterbalance in E1? Also, might children’s responses to the Endorse questions have biased their subsequent responses to the Ask questions?

b. Each experiment tests a different age range, but there’s really no explanation why. From E1 to E2 3s and 9-12s are dropped. Since 9-12s are not assessed separately in E1, there’s no obvious reason why they should be eliminated from E2. And, quite confusingly, 4s were dropped from E3. This is confusing because 4s did provide interesting data in E2 and because the authors state that they “opted” not to include younger participants based on the “analyses of these data” but provide no further explanation/justification.

c. In E2, the question in the History Phase (“What do you think it’s a picture of? A rabbit or a puppy?”) (PG 22) seems to be identical to the Endorse question from E1. But, then E2 also includes an endorse question. Was it the same question or did it change?

d. In E2 the response to the History Phase is included with the “learning” DVs but it is not included with those for E1 or E3. The authors explain that, at least for E3, the decision to separate History was based on one informant having visual access (PG 27). But, that is also true in E2 where History is not separated.

e. The abrupt change in analysis from E1 (regression) to E2 and E3 (t-test) was confusing given how similar the designs were across the 3 experiments. For overall consistency and for ease in comparison across studies, I would recommend that the authors use the same analysis throughout.

f. Age is treated as a continuous variable in the main analysis for E1, but as categorical for the main analysis in E2 and E3.

g. Amanda is frequently favored over Emily, sometimes by a wide margin (it appears that informant identity might be a stronger effect than condition). The authors state that they will discuss the differences but never really do.

h. Comparisons to chance are provided for E2 and E3, but not E1. Given that many of the E1 proportions hover around 50%, and may not differ from chance, it is important to know if random guessing could account for children’s pattern of performance.

i. “Directional” tests seem to only be used when the analysis yields a p-value close to .05 which makes the decision seem convenient (to achieve a significant effect below the p = .05 cutoff) rather than theoretical.

2) There are some places where clarification would be helpful. For example:

a. E1 states that both models answered “confidently” in the test phase but the example script provided indicates the model say “I think…” (PG 13) which is a cure to hesitancy, not confidence. Given that both informants do this it’s probably not meaningful for the results, but it still might be worth clarifying in the E1 discussion.

b. The authors choose to include children who fail the manipulation check (which amounts to 10-26% depending on E and question) in the analyses. “Being conservative” and “maintaining representativeness” are given as supporting reasons in E1, but I worry about these. In terms of representativeness, E1 benefits from a huge sample size (at least by developmental research standards) and I would think it could absorb the loss of some participants without threatening representativeness. If not, the authors could analyze the “failed” sub-sample and list exactly how they erode representativeness. The “conservativeness” argument sounds honorable, but given how close children’s performance is to 50/50 in so many cases I think this approach is not fair to the data. And, logically, how can children who fail to correctly identify confident and hesitant informants be expected to use confidence/hesitancy to later distinguish between the testimony of those informants?

c. In E3, the authors report that 7-8s “preferred to learn” (PG 27) but the only time this age group differed was during the history (and not the “future learning”) phase.

d. Is there a typo after “Trial Type” (PG 27) … should history be deleted?

e. I really appreciated the paragraph (PG 34-35) detailing the real world challenge of detecting and using calibration. One additional point the authors might consider including in this paragraph is that identifying a well-calibrated individual in context often means knowing whether the informant is accurate or not. Paradoxically however, if the learner knew the accurate information she would not need an informant (i.e., there’d be no need for learning).

f. It would be helpful if the authors enlisted specific findings and citations to clarify support the 3-stage progression for understanding confidence and hesitancy (PG 35-36). In particular, it was not clear what the best support for Stage 3 was. Perhaps the authors want to suggest that 7 to 8-yr-olds response to the History question in E3 supports Stage 3, but this age group does not differentiate between hesitant informants on most of the DVs in E3 and the abstract describes this age group as “insensitive to calibration”. Or, perhaps they see the Brosseau-Liard et al. results as supporting Stage 3, but then it becomes difficult to understand the timeline for when children might progress through the stages.

g. Given its relevance to this paper, the Huh et al. paper should be cited and discussed earlier.

6. PLOS authors have the option to publish the peer review history of their article (what does this mean?). If published, this will include your full peer review and any attached files.

Reviewer #1: No

---

## [Author Response · Author response to Decision Letter 0]

28 Nov 2019

Response to Reviewers

Reviewer #1: Below is a review of submission PONE-D-19-18550:

Children’s understanding of when a person’s confidence and hesitancy is a cue to their credibility

This paper reports on findings from three experiments looking at children’s understanding of calibration, the match between an informant’s confidence/hesitancy and knowledge. E1 asks whether 3 to 12-year-olds prefer to learn from “well-calibrated” informants (those who are confident when knowledgeable and hesitant when not), and whether children view well-calibrated informants as “smarter”. The remaining experiments follow-up E1 by equating the informant’s level of confidence (E2) and hesitancy (E3). Findings suggest that by age 4 or 5 children differentiate between well-calibrated from mis-calibrated informants (E1), provided those informants had a history of being confident (E2) rather than hesitant (E3). Overall the article is well-written, the findings are appropriate for the journal, and the topic is likely to be of interest to the journal’s readership. Below are several suggestions intended to help make future drafts of the paper stronger.

1) There are some inconsistencies across the 3 experiments, many of which are not clearly motivated or explained. For example:

a. The Endorse and Ask questions are counterbalance in E2 and E3, but not E1. Why not counterbalance in E1? Also, might children’s responses to the Endorse questions have biased their subsequent responses to the Ask questions?

RESPONSE: Thank you for pointing out that we should have clarified the rationale for this particular aspect of our design. We have now done so on p. 14. In short, for Experiment 1, endorse trials involve the participants hearing the two models confidently provide labels for objects they can see, whereas Ask trials do not involve the models saying anything at all. We reasoned that before deciding who to learn from it was important that participants have evidence (from the model speaking in the first Endorse trial) that the previously hesitant individual is not always hesitant (i.e., she is only hesitant when she has not seen the object being labeled). Therefore, it was critical for the Endorse Trials to proceed the Ask Trials in Experiment 1. For the subsequent experiments, this was not an issue given that the models were both confident or both hesitant during the History Phase (rather than one being confident and one being hesitant), so and we chose to counterbalance the Ask and Endorse trials along with the other variables (as discussed on p. 23). 

b. Each experiment tests a different age range, but there’s really no explanation why. From E1 to E2 3s and 9-12s are dropped. Since 9-12s are not assessed separately in E1, there’s no obvious reason why they should be eliminated from E2. 

RESPONSE: Our age analyses for E1 revealed remarkably little developmental change from the preschool period through age 12. Post-hoc analyses on the age groups revealed that children understood over-confidence calibration by ages 5 & 6 and that there were no age differences between the 5&6-year-old group and the older children. Given the lack of developmental change above age 6 in E1, we felt including children above age 8 in E2 and E3 was of little interest to understanding the developmental picture. E1 suggested the “developmental action”, so to speak, was in the younger ages. The mean age of our youngest group in Experiment 1 was 3.93 years so we reduced our sample size and narrowed our age range to include children 4 to 8 years. We have added a statement to the participants section of E2 (p. 21) to clarify the rationale for selecting children ages 4-8. We also added a statement (p. 26) acknowledging that since our results revealed that even 4-year-olds (our youngest group) are sensitive to confidence calibration future work would be needed to identify precisely when in development children first become sensitive. 

And, quite confusingly, 4s were dropped from E3. This is confusing because 4s did provide interesting data in E2 and because the authors state that they “opted” not to include younger participants based on the “analyses of these data” but provide no further explanation/justification.

RESPONSE: We thank you for pointing out that our statement justifying our sample for E3 needs clarification. We hypothesized that this would be a harder inference to make so we began by testing the older children and analyzed those results before deciding whether to include younger participants. We have added this clarification on p. 29. That is, because the analyses revealed that neither the 5/6s or the 7/8s showed a significant preference to learn from the calibrated model there was no reason to believe that younger children would. 

c. In E2, the question in the History Phase (“What do you think it’s a picture of? A rabbit or a puppy?”) (PG 22) seems to be identical to the Endorse question from E1. But, then E2 also includes an endorse question. Was it the same question or did it change?

RESPONSE: The Endorse Questions and Ask Questions were the same for all three Experiments. The Endorse questions were the same type of questions as in the History Phases but unlike the History Phases which involves familiar animals and labels (e.g. rabbit or puppy), the Endorse questions always involve the third party, John, showing the models pictures of ‘strange’ (unfamiliar) animals and the models providing unfamiliar labels (e.g. toma or modi) to deduce whether children prefer to learn new object labels from one model over the other. We’ve added a clarification to the methods of E2 where we first introduce the History trials as a D.V. (p. 22-23). 

d. In E2 the response to the History Phase is included with the “learning” DVs but it is not included with those for E1 or E3. The authors explain that, at least for E3, the decision to separate History was based on one informant having visual access (PG 27). But, that is also true in E2 where History is not separated.

RESPONSE: Apologies for the confusion here and again thank you for pointing out what aspects of our rationale need clarifying. In E1 it is essential that participants are told that BOTH models are wrong during the history trials (to create one well-calibrated, hesitant and wrong, and one poorly calibrated model—confident and wrong). As a result, we did not expect them to learn from either model during the history trials and so did not force them to pick between 2 poor (wrong) choices in case their arbitrary decision to favor one model unnecessarily influenced their later responses. For E2 and E3, in contrast, there is a correct answer for who they should trust during the History trials so we wanted to see who they learned from here. We’ve added a clarification to the method section of E2 where we first introduce the History trials as a D.V. (p. 23). 

Further, we report their performance on the history trials in both E2 and E3 but suspect the confusion arose from how we previously presented the results. That is, because the correct answer in E2 on the history trials is also the well-calibrated model we could combine the DVs. Whereas, the correct answer for the history trials in E3 is NOT the calibrated model (but rather the one who is informed). Since the direction of the predicted direction of the preference should switch between History and later Learning Trials it is not appropriate to combine them as in E2. We think the new results sections for both E2 and E3 clarifies this confusion by separating the history trials for both E2 and E3. 

e. The abrupt change in analysis from E1 (regression) to E2 and E3 (t-test) was confusing given how similar the designs were across the 3 experiments. For overall consistency and for ease in comparison across studies, I would recommend that the authors use the same analysis throughout.

RESPONSE: Thank you for this suggestion. We initially planned to analyzed all 3 experiments by age-groups with ANOVAs and t-tests (which seem to be more widely used and understood by some readerships). The decision to use regression for E1 stemmed primarily from the need to control for the unexpected bias that emerged in E1 to favor one of the two models/informants. Nonetheless, we agree that using the same analyses for all 3 experiments will make for easier comparisons, so we now provide regression analyses for all 3 experiments, along with additional analyses treating age categorically like we did in the previous write-up of E1. 

f. Age is treated as a continuous variable in the main analysis for E1, but as categorical for the main analysis in E2 and E3.

RESPONSE: As noted above, we now provide the same analyses for all three experiments. 

g. Amanda is frequently favored over Emily, sometimes by a wide margin (it appears that informant identity might be a stronger effect than condition). The authors state that they will discuss the differences but never really do.

RESPONSE: Thanks for pointing this out. Previous literature has noted that children sometimes show preferences to learn based on visual attractiveness and this can even trump more direct cues to one’s credibility such as prior accuracy. We have added a discussion of the possible implications of this model bias on our results to the General Discussion on p. 41. 

h. Comparisons to chance are provided for E2 and E3, but not E1. Given that many of the E1 proportions hover around 50%, and may not differ from chance, it is important to know if random guessing could account for children’s pattern of performance.

RESPONSE: Unfortunately, t-tests cannot account for the unexpected model bias so we used regression analyses to control for the model bias. When the model bias is controlled for, significant learning preference emerge that cannot be explained by chance (i.e., random guessing). We now report regression analyses for all 3 Experiments for consistency. Specifically, for Experiment 1 (p. 16, Table 2, Model 2, Condition, p = .006), the preference to learn from the model who was previously confident when uninformed was significantly different from chance (i.e., reduced). That is, as expected, children avoided learning from a currently confident source if her previous confidence was unjustified (i.e., she was confident when uninformed). 

i. “Directional” tests seem to only be used when the analysis yields a p-value close to .05 which makes the decision seem convenient (to achieve a significant effect below the p = .05 cutoff) rather than theoretical.

RESPONSE: An alpha of .05 is used throughout. In 2 places we report a directional/one-tailed p value = .029 (over a p = .058, two tailed) simply to remind our reader that there is an a priori hypotheses about the direction of the learning preference making a two-tailed ‘marginal’ finding misleading. With few exceptions, where noted, all of our analyses are based on a priori hypotheses about the direction of the learning preferences therefore theoretically all of the analyses are supported by directional tests—We simply chose to report the two-tailed values throughout and only remind the reader of the directional nature of the hypotheses for any results that might have appeared marginal. If you prefer, for consistency, we would be happy to change all of the p values to directional tests where applicable. Additionally, these directional t-tests are now only reported in the follow-up analyses to confirm the results from the main regression analyses. 

2) There are some places where clarification would be helpful. For example:

a. E1 states that both models answered “confidently” in the test phase but the example script provided indicates the model say “I think…” (PG 13) which is a cure to hesitancy, not confidence. Given that both informants do this it’s probably not meaningful for the results, but it still might be worth clarifying in the E1 discussion.

RESPONSE: Thank you for this suggestion. This is a really interesting point to discuss. We have added a brief clarification on page 13 pointing out that even though the word ‘think’ is sometimes used to convey hesitancy it can also be used when highlighting that one’s statement does, or might, conflict with another’s response (as in the current context). We also noted that although Moore et al (1989) showed that by 4 years of age children prefer to trust a source that uses the term “I know” over “I think”, this does not necessarily imply that they always equate this with hesitancy (only that it is less convincing than know). Our results suggest that young children do not appear confused or deterred by the otherwise confident speaker using this language—perhaps as you wisely point out because both models use the word and there are other cues available to them to infer confidence and hesitancy. 

b. The authors choose to include children who fail the manipulation check (which amounts to 10-26% depending on E and question) in the analyses. “Being conservative” and “maintaining representativeness” are given as supporting reasons in E1, but I worry about these. In terms of representativeness, E1 benefits from a huge sample size (at least by developmental research standards) and I would think it could absorb the loss of some participants without threatening representativeness. If not, the authors could analyze the “failed” sub-sample and list exactly how they erode representativeness. The “conservativeness” argument sounds honorable, but given how close children’s performance is to 50/50 in so many cases I think this approach is not fair to the data. And, logically, how can children who fail to correctly identify confident and hesitant informants be expected to use confidence/hesitancy to later distinguish between the testimony of those informants?

RESPONSE: Based on other findings in the literature it appears that children can fail to correctly answer explicit questions, at the end of an experiment where it is possible they are also fatigued, despite showing evidence of understanding in the earlier questions. Nonetheless we completely understand the concern here, so for the sake of being thorough and transparent we now report all analyses with and without exclusions. The same pattern of results were obtained in all 3 experiments whether we exclude or don’t exclude participants who failed the explicit manipulation check. Therefore, we report the full sample results in the main analyses and the analyses with exclusions in the supplemental materials. 

c. In E3, the authors report that 7-8s “preferred to learn” (PG 27) but the only time this age group differed was during the history (and not the “future learning”) phase.

RESPONSE: Yes, that is correct—the learning preference only applied to the History Phase. This should be clearer in the revised manuscript which separates the analyses for history trials and future learning phases. 

d. Is there a typo after “Trial Type” (PG 27) … should history be deleted?

RESPONSE: Yes, thank you for spotting this typo. This sentence no longer exists in the revised results section. 

e. I really appreciated the paragraph (PG 34-35) detailing the real world challenge of detecting and using calibration. One additional point the authors might consider including in this paragraph is that identifying a well-calibrated individual in context often means knowing whether the informant is accurate or not. Paradoxically however, if the learner knew the accurate information she would not need an informant (i.e., there’d be no need for learning).

RESPONSE: Yes, good point. I think the important distinction with calibration tracking is that it is useful when one’s current accuracy is unknown (in a given situation) but one’s prior track-record or history of accuracy is (and how well their prior accuracy is calibrated with their prior confidence/hesitancy). This memory component is important. We have expanded on this discussion on p. 40.

f. It would be helpful if the authors enlisted specific findings and citations to clarify support the 3-stage progression for understanding confidence and hesitancy (PG 35-36). In particular, it was not clear what the best support for Stage 3 was. Perhaps the authors want to suggest that 7 to 8-yr-olds response to the History question in E3 supports Stage 3, but this age group does not differentiate between hesitant informants on most of the DVs in E3 and the abstract describes this age group as “insensitive to calibration”. Or, perhaps they see the Brosseau-Liard et al. results as supporting Stage 3, but then it becomes difficult to understand the timeline for when children might progress through the stages.

RESPONSE: Thanks for pointing out that this needs further clarification. We specify that we are only cautiously offering this 3-stage model precisely because we can’t enlist many specific findings to support it. Essentially, we are using our data here as the first evidence against the 2-stage model implied by earlier work (e.g. Tenney et al), whereby one goes from not understanding calibration to understanding calibration). That is, we find evidence against a simple 2-stage model implying the existence of at least one additional stage (i.e. a 3-stage model). As you point out, the children in our experiments appear to have not reached Stage 3, suggesting Stage 3 is a later developmental accomplishment (since adults clearly understand that not previously hesitant people are unreliable sources of information). We will be an interesting line for future research. We have expanded our discussion on this on pp. 42-43.

g. Given its relevance to this paper, the Huh et al. paper should be cited and discussed earlier.

RESPONSE: Although relevant to our discussion, this paper did not exist in advance of our experiments and therefore did not impact our hypotheses or design. It was published after our data were already collected, analyzed, and presented at conferences). As a result, it felt misleading or misrepresentative to discuss that work in the introduction to our paper as it did not inform our research hypotheses. We have added dates of data collection statements to our method sections. Still, since the article became available before we published our results we felt it was worth including in the discussion so readers can compare and contrast.

---

## [Decision Letter · Decision Letter 1]

12 Dec 2019

Children's understanding of when a person's confidence and hesitancy is a cue to their credibility

PONE-D-19-18550R1

Dear Dr. Severson,

We are pleased to inform you that your manuscript has been judged scientifically suitable for publication and will be formally accepted for publication once it complies with all outstanding technical requirements.

With kind regards,

Valerio Capraro

Academic Editor

PLOS ONE

Additional Editor Comments (optional):

Reviewers' comments:

Reviewer's Responses to Questions

**Comments to the Author**

1. If the authors have adequately addressed your comments raised in a previous round of review and you feel that this manuscript is now acceptable for publication, you may indicate that here to bypass the “Comments to the Author” section, enter your conflict of interest statement in the “Confidential to Editor” section, and submit your "Accept" recommendation.

Reviewer #1: All comments have been addressed

2. Is the manuscript technically sound, and do the data support the conclusions?

Reviewer #1: Yes

3. Has the statistical analysis been performed appropriately and rigorously? 

Reviewer #1: Yes

4. Have the authors made all data underlying the findings in their manuscript fully available?

Reviewer #1: Yes

5. Is the manuscript presented in an intelligible fashion and written in standard English?

Reviewer #1: Yes

6. Review Comments to the Author

Reviewer #1: Overall, I am impressed with the thoroughness of the revision and the authors’ attentiveness to my previous comments. Originally, I was most concerned with the model bias (favoring Amanda), what appeared to be near-chance performance, and some unexplained methodological and analytical choices. The authors have thoughtfully responded to each of these, providing important details that lessen my concerns. I am especially appreciative of the authors’ efforts to re-analyze the data to include both the sample with and the sample without exclusions. I’m satisfied with the revision and have no further comments.

7. PLOS authors have the option to publish the peer review history of their article (what does this mean?). If published, this will include your full peer review and any attached files.

Reviewer #1: No

---

## [Editor Report · Acceptance letter]

27 Dec 2019

PONE-D-19-18550R1 

Children's understanding of when a person's confidence and hesitancy is a cue to their credibility 

Dear Dr. Severson:

I am pleased to inform you that your manuscript has been deemed suitable for publication in PLOS ONE. Congratulations! Your manuscript is now with our production department. 

With kind regards,

on behalf of

Dr. Valerio Capraro 

Academic Editor

PLOS ONE